# A Bi-FPN-Based Encoder–Decoder Model for Lung Nodule Image Segmentation

**DOI:** 10.3390/diagnostics13081406

**Published:** 2023-04-13

**Authors:** Chandra Sekhara Rao Annavarapu, Samson Anosh Babu Parisapogu, Nikhil Varma Keetha, Praveen Kumar Donta, Gurindapalli Rajita

**Affiliations:** 1Indian Institute of Technology (Indian School of Mines), Dhanbad 826004, India; acsrao@iitism.ac.in (C.S.R.A.); keethanikhil@gmail.com (N.V.K.); 2Krishna Chaitanya Institute of Technology and Sciences, Markapur 523316, India; 3Distributed Systems Group, TU Wien, 1040 Vienna, Austria; 4Department of ECE, GIET University, Gunupur 765022, India; chanduraji@gmail.com

**Keywords:** segmentation, deep learning, computed tomography, medical image analysis

## Abstract

Early detection and analysis of lung cancer involve a precise and efficient lung nodule segmentation in computed tomography (CT) images. However, the anonymous shapes, visual features, and surroundings of the nodules as observed in the CT images pose a challenging and critical problem to the robust segmentation of lung nodules. This article proposes a resource-efficient model architecture: an end-to-end deep learning approach for lung nodule segmentation. It incorporates a Bi-FPN (bidirectional feature network) between an encoder and a decoder architecture. Furthermore, it uses the Mish activation function and class weights of masks with the aim of enhancing the efficiency of the segmentation. The proposed model was extensively trained and evaluated on the publicly available LUNA-16 dataset consisting of 1186 lung nodules. To increase the probability of the suitable class of each voxel in the mask, a weighted binary cross-entropy loss of each sample of training was utilized as network training parameter. Moreover, on the account of further evaluation of robustness, the proposed model was evaluated on the QIN Lung CT dataset. The results of the evaluation show that the proposed architecture outperforms existing deep learning models such as U-Net with a Dice Similarity Coefficient of 82.82% and 81.66% on both datasets.

## 1. Introduction

According to data released by the World Health Organization (WHO) on 3 February 2020, cancer is one of the leading causes of premature death in 134 of 183 countries. It has been observed that, especially in 2018, most of the prominent cancer deaths are due to lung cancer (1.76 million deaths). Detection and analysis of the lung nodules at an early stage facilitate efficient treatment and drastically improve a patient’s chance of survival [1]. CT scans are a widely used and highly accurate format for the purpose of screening and analyzing lung nodules, especially in differentiating the nodules from other structures. Moreover, the precise segmentation of these nodules is critical, considering the heterogeneity of the size, texture, location, and shape of the nodules, and the fact that their intensity may differ within the borders [2]. There are various types of lung nodules as observed in Figure 1 such as adhesion-type (juxtapleural and juxta-vascular), isolated, cavitary, calcified, small, and ground-glass opacity (GGO) nodules [3].

Another challenge lies in the segmentation of lung nodules, which is found in the case of nodules with small diameter and intensity comparable to that of the surrounding noise, which thereby hinders the down-sampling potential of the segmentation network, where the network cannot extract more in-depth semantic network features [4]. It significantly impacts the accuracy of the extraction of feature maps of large nodules. Based on these reasons, a robust segmentation network is necessary to accommodate the large-scale nodule (various types) problem.

Recently, convolutional neural networks (CNN) have become the mainstream architecture in the field of computer vision. One such architecture, the U-Net, is an encoder–decoder-like CNN architecture, which has shown exceptional results in segmentation of biomedical images [5]. Many modified U-Net architectures have achieved significant results in different domains of biomedical imaging. However, CNN architectures that are implemented for the task of lung nodule segmentation are still immature. Therefore, the development of advanced architectures dealing with the shortcomings of previous architectures is essential.

To deal with the challenges of efficient feature extraction and adaptation to heterogeneity of lung nodules, this paper proposes a modified U-Net architecture with a weighted bidirectional feature network (U-Det), which is appropriate for the segmentation of many forms of lung nodules. Figure 2 illustrates the pipeline of the proposed model.

### Contributions

The list of the following elements provides technical contributions of this research:The proposed U-Det model uses a bidirectional feature network (Bi-FPN), which functions as a feature enricher, integrating multi-scale feature fusion for efficient feature extraction.Applying a data augmentation technique to deal with the small-size dataset prevents the model from over-fitting and provides better segmentation results.Implementing the Mish activation function, due to its strong regularization effects, provides enhanced model training and segmentation efficiency.Comparing the proposed U-Det model to the existing U-Net to the existing U-Net shows its high segmentation performance on small nodules and various categories of other pulmonary nodules.

## 2. Background and Related Work

This section describes the existing conventional and machine-learning-based segmentation approaches for lung nodule segmentation.

### 2.1. Conventional Approaches

Many conventional approaches, such as morphological methods, region-growing processes, energy-based optimization techniques, and machine learning methods, were proposed for lung nodule segmentation in the literature. In morphological methods, morphology-based operations and those based on shape hypothesis were applied to isolate lung nodules by selecting the connected region [6]. However, the isolation of lung nodules using morphological operations did not perform efficiently [7]. Following that, region-growing methods were proposed to improve lung nodule segmentation but it was observed that these methods were unable to segment juxta-vascular and juxtapleural nodules and were only well suited to isolate calcified nodules [3]. In this case, for region-growing methods, the convergence condition and irregular-shaped nodules create difficulty due to the breach of the shape hypothesis. Similarly, energy based-optimization methods were also proposed for lung nodule segmentation. These methods consist of a level set function for the characterization of the image, and an energy function to typically turn the segmentation task into an energy minimization problem [8]. However, juxtapleural nodules and low contrast nodules such as GGO nodules drastically affect the improvement of lung nodule segmentation.

### 2.2. Machine-Learning-Based Approaches

In the past decade, many machine-learning–based methods were proposed for lung nodule segmentation [9]. An example of one such approach is the hybrid model for classifying the lung nodules with high-level feature maps proposed by Lu et al. [10]. Further, methods based on modified support vector machines were suggested to detect small lung nodules in 3D CT scans [11,12]. Down the line, researchers also developed 3D large-scale nodule segmentation techniques based on the Hessian strategy in combination with neural networks [13]. Similarly, segmentation approaches based on multi-phase models [14], unsupervised k-means clustering, and level set algorithms [15] were also proposed.

Among the recently developed deep learning-based approaches for segmentation, CNN is a multi-layered neural network that learns to map original image files and corresponding labels hierarchically, and to change the segmentation task into the classification of voxels [16,17]. For instance, Wang et al. introduced a multi-view CNN (MV-CNN) for nodule segmentation [18]. Further, Huang et al. developed a fast and fully automated detection and segmentation approach for pulmonary nodules in thoracic CT scans using deep convolutional neural networks [19]. Moreover, the authors Subham and Raman developed a lung nodule segmentation approach Using 3-dimensional convolutional neural networks [20]. A good review is also available on pulmonary nodule detection and diagnosis using deep learning applications in computed tomography images [21]. Moreover, Havaei et al. in [22] proposed a cascaded variant of CNN (Cascaded-CNN) for brain tumor segmentation. Further extending the use of CNN to lung nodule segmentation, Shen et al. proposed a multi-crop CNN (MC-CNN) to extract salient nodule features and classify malignancy [16]. Furthermore, Wang et al. in [18] introduced a multi-view deep CNN (MV-DCNN), and Sun et al. in [23] introduced a three multi-channel region of interest (RoI)-based CNN (MCROI-CNN) for lung nodule segmentation and displayed the effectiveness of CNN over existing computer-aided diagnosis systems. Further expanding work on CNN-based architectures, Zhao et al. have advocated an enhanced pyramid deconvolution network for increasing performance on lung nodule segmentation which blends low-level fine-grained characteristics with high-level functional characteristics [24].

On the other hand, fully convolutional networks [25] are a different approach for CT image segmentation. For example, the architectures 2D U-Net and 3D U-Net, proposed by Ronnerberger et al. and Ciccek et al., respectively, are better adapted to biomedical imaging [5,26]. The fully convolutional network U-Net (FCN-UNET) architecture is a convolutional network architecture used for fast and precise segmentation of images. Similarly, the central focused CNN (CF-CNN) proposed by Wang et al. is a data-driven method without involving the shape hypothesis. It showed strong performance for the segmentation of juxtapleural nodules [27]. Moreover, Cao et al. recently proposed incorporating intensity features into the CNN architecture by implementing a dual-branch residual network (DB-ResNet) [28].

Segmentation 2D convolutional U-Network (SegU-Net), a U-Net-based architecture, has also been suggested for lung nodule segmentation [29]. For pre-processing, Wang et al. and Cao et al. implemented weighted sampling of training data to deal with the comparatively smaller size of the Lung Image Database Consortium and Image Database Resource Initiative (LIDC-IDRI) dataset [27,28]. The input slice of a CT scan is cropped to a smaller size with a random weighted sampling strategy to increase the size of the training dataset in the abovementioned method. Recently, Singadkar et al. proposed another approach toward lung nodule segmentation based on a deep deconvolutional residual network (DDRN) [30]. In DDRN, the approach was based on an encoder–decoder architecture with one-directional long-skip connections. Moreover, Wen-Fan Chen et al. developed a residual-dense-attention (RDA) U-Net network architecture for hepatocellular carcinoma segmentation [31].

## 3. Proposed Method

This section explains the following three phases of the proposed model: (1) data augmentation, (2) model architecture, and (3) training and post-processing. Figure 3 visualizes the proposed U-Det model architecture.

### 3.1. Data Augmentation

Due to the constraints of much available training data in medical image segmentation, data augmentation helps in preventing the model from over-fitting. It also improves the generalization capability of the network on data outside the training set. Along with that, it plays a vital role in building robust deep learning pipelines [32,33]. A data augmentation strategy instead of a sampling strategy was applied to input CT images of size 512×512 for maintaining the same input size in the proposed model. Data augmentation methods applied in the proposed network are scale, flip, shift, rotate, salt and pepper noise, and elastic deformations [34]. By applying these small transformations to images during training, variety was created in the training dataset and robustness of the proposed model was improved.

### 3.2. Model Architecture

The architecture of the proposed model is an end-to-end deep learning approach for lung nodule segmentation, inspired by the encoder–decoder backbone of U-Net, and the feature enricher Bi-FPN. The proposed U-Det architecture is designed to take a 512×512 image as an input and a 512×512 mask as output. It consists of three sections: the contraction, Bi-FPN, and the expansion section of depth 5, which behave similarly to an encoder, feature enricher, and decoder. Table 1 shows the corresponding layers of the model, along with their respective parameters.

Initially, the contraction section takes the augmented CT image, a slice of the CT scan. Here, the repeated application of two 3×3 convolutions (with the ‘same’ padding) is applied using Equation (Equation 1)
(1)$$C\left[m,n\right]=\left(I\times k\right)\left[m,n\right]=\sum_{i}\sum_{j}k\left[i,j\right].I\left[m-i,n-j\right],$$
where C[m,n] represents the kernel convolution, *I* and *k* denote the input image and kernel, respectively, and ‘*m*’ and ‘*n*’ represent the number of ‘training samples’ and ‘features’, respectively. Further, each convolution is followed by a recently proposed non-linear Mish activation function [35], to perform strong regularization during the forward and backward pass of the model, the formula of which is shown in Equation (Equation 2)
(2)f(x)=x.tanh(ω(x)),
where ω(x) is the Softplus activation function given by $ln(1+e^{x})$.

Mish implements a self-gating function, in which the input given to the gate is a scalar. The property of self-gating helps in replacing the activation functions (point-wise functions) such as rectified linear unit (ReLU). Here, the input of the gating function is a scalar with no requirement of modifying network parameters. The properties of Mish, being above-unbounded, below-bounded, non-monotonic, and smooth, play a vital part in maximizing neural network outcomes. Hence, Mish enables considerable time improvements during the forward and backward pass on Graphics Processing Unit (GPU) inference when Compute Unified Device Architecture (CUDA) is enabled and improves the efficiency of the model.

Further, a 2×2 max-pooling operation of stride ‘2’ is applied for down-sampling of the input image features. Here, at depth 4, a dropout layer with a dropout factor of 0.5 is used for the regularization of the model. The feature maps of the corresponding five depths of the contraction path are then fed as input to the Bi-FPN, as illustrated in Figure 3.

For efficient feature extraction, Bi-FPN implemented in the proposed method is based on conventional top-down Feature Pyramid Networks (FPN) [36]. It infuses efficient bidirectional cross-scale connections and weighted feature fusion into the model. In the Bi-FPN, multi-scale feature fusion aims to fuse features at different resolutions to obtain efficient feature extractions. In contrast, the one-way flow of information inherently limits conventional top-down FPN. Thus, Bi-FPN does not consist of nodes with only one input edge because if a node has only one input with no feature fusion, it will contribute less to the feature network to infuse different features. Moreover, the Bi-FPN has one top-down and one bottom-up path, thereby allowing the bidirectional flow of features from one depth to another in the feature network.

The Bi-FPN also incorporates additional weight for each input during feature fusion, thereby learning the importance of a particular input feature. In the Bi-FPN, dynamic learning behavior and accurate fast normalized fusion (one of the methods of incorporating weights during feature fusion) are implemented. Moreover, for improved efficiency, depth-wise separable convolution followed by batch normalization and non-linear ReLU activation function are implemented. Through the bidirectional cross-scale connections, the Bi-FPN enriches the feature maps at each depth of the network and provides an efficient fusion of features across various depths of the encoder–decoder architecture.

Incorporating a bidirectional feature network aims to improve the efficiency of feature extraction at each level of the backbone architecture and enrich the feature vectors, thus allowing a fusion of lower-level fine-grained and higher-level semantic features. As illustrated in Figure 3, the inputs of the Bi-FPN are the feature maps of the corresponding five depths of the contraction path of the backbone architecture. Then, the outputs of Bi-FPN are fed into the expansion path of the backbone network.

Inside the expansion section, the outputs of Bi-FPN are each combined with a decoder architecture to obtain a combination of lower-level fine-grained features with high-level semantic features. Each step in the expansion path consists of an up-sampling of the feature map followed by a 2×2 convolution (up-convolution), which halves the number of feature channels at each depth. The feature vectors obtained after up-sampling are concatenated with the corresponding feature vectors from the Bi-FPN. The concatenation operation is followed by two 3×3 convolutions (with the ‘same’ padding), and each is followed by the Mish activation function. In the final layer of the expansion section, the obtained 512×512×64 feature map undergoes two 3 × 3 convolutions, which are again followed by the Mish activation function. Then a 1×1 convolution block and a sigmoid activation function are applied as the final operation. Thus, they obtain logits corresponding to the mask of the input CT image of 512×512.

### 3.3. Training and Post-Processing

The network training aims to increase the probability of the suitable class of each voxel in the mask. In respect to that, a weighted binary cross-entropy loss of each sample for training was utilized. The positive pixels, by the ratio of negative-to-positive voxels, in the training set were weighted to implement weighted binary cross-entropy. Since the size of the positive class in a lung nodule mask is relatively smaller compared to that of the negative class, the training set’s class weight is positive, thereby increasing the punishment for getting a positive value wrong. So the network will learn to be less biased toward outputting negative voxels due to the class imbalance in the masks. The weighted binary cross-entropy loss is formulated as follows:(3)$$Loss=\frac{-1}{N}\sum_{i=1}^{N}\left[\omega_{p}\times y_{i}\log\hat{y_{i}}+\left(1-y_{i}\right)\log\left(1-\hat{y_{i}}\right)\right],$$
where, *N* represents the number of samples, $\omega_{p}$ represents the positive prediction weights, $y_{i}$ represents the ground truth, and $\hat{y_{i}}$ indicates the prediction of the U-Det model.In the training approach, K-fold cross-validation [37] was utilized to obtain an accurate measure of the generalizing capability of the proposed model. Furthermore, to deal with the generation of augmented training CT images and corresponding ground truths, generators were implemented to augment input images and the generation of corresponding ground truth labels. For model optimization, the Adam model optimization algorithm [38], which updates network weights requiring little hyper-parameter fine-tuning, was utilized with the following hyper-parameters: the initial learning rate is 0.0001, Beta_1 = 0.99, Beta_2 = 0.999, and the decay rate is 1 × 10${}^{-6}$. Moreover, a batch size of two samples was chosen based on the memory size of the GPU for training the model. Further, the early stopping training strategy [39] was applied to prevent over-fitting. Additionally, the strategy of reducing the learning rate of the optimizer once the model’s performance reaches a plateau was utilized. In the post-processing phase, the proposed model was designed to save the final obtained masks in a raw, metal (.mhd) format, storing volumetric data such as CT scans.

## 4. Data and Experiments

This section explains the details regarding the dataset, implementation, and evaluation metrics.

### 4.1. Data

For the experimentation and training of the proposed model, the approach utilized the publicly available dataset of the Lung Nodule Analysis 2016 (LUNA16) grand challenge [40,41,42], which is derived from the public LIDC-IDRI dataset [43,44]. In total, 888 CT scans are included in the dataset. Figure 4 illustrates the histogram of the nodule amount and diameter of nodules for the LUNA16 dataset. In addition to the LUNA16 dataset, the Quantitative Imaging Network (QIN) [45] Lung CT Segmentation dataset was used to evaluate the effectiveness of the proposed model. The QIN dataset contains 41 CT scans of non-small-cell lung cancer collected from different sources, such as LIDC, Reference Image Database to Evaluate Therapy Response (RIDER), Stanford University Medical Center, the Moffitt Cancer Center, and the Columbia University Phantom [46,47,48]. This dataset contains a total of 52 lung CT nodules across the 41 CT scans.

In the pre-processing phase, the CT images and ground truth masks were generated from CT scans, utilizing the Kaggle Data Science Bowl 2017 Competition based on annotations and a mask creation algorithm [49]. By using this method we obtained a total of 1166 CT images with corresponding ground truth masks. Further, it was partitioned into a training and a test subset containing 922 and 244 images, respectively, and we applied K-fold cross-validation of 4-folds during model training. The two subsets showed identical statistical distribution in their clinical characteristics, as depicted in Table 2. Furthermore, the same pre-processing strategy was used on the QIN dataset with the provided lung nodule locations to generate 156 CT images with corresponding ground truth masks.

### 4.2. Evaluation Metrics

The Dice Similarity Coefficient (DSC) was used as the key evaluation metric for assessing the segmentation performance of the U-Det model. It is a commonly used metric to calculate the difference between the outcomes of two segmentations [22,50]. In addition to the abovementioned metrics, the sensitivity (SEN) and positive predictive value (PPV) were used as additional evaluation metrics, and are formulated as follows
(4)$$DSC=\frac{2\times V(Gt\cap Sv)}{V(Gt)+V(Sv)},$$
(5)$$SEN=\frac{V(Gt\cap Sv)}{V(Gt)},$$
(6)$$PPV=\frac{V(Gt\cap Sv)}{V(Sv)},$$
where “*Gt*” represents the ground truth labels, and “*Sv*” represents the segmentation results of the proposed model. Here, the volume size measured in voxel units is represented by “*V*”.

### 4.3. Implementation Details

The experimental implementation was performed using TensorFlow (Version 2.1) deep learning framework (GPU version), Python 3.6 was the language used for coding, and CUDA 10.2 was applied for accelerated training. The experiment was carried out using Google cloud platform on a virtual instance equipped with 4 vCPUs, 15GB memory, and an SSD drive of 500 GB. During the model’s training, acceleration was performed using NVIDIA Tesla T4 GPU (14 GB video memory), and it needed about 8 h of training (20 epochs) to converge. The following link provides the source code of the implementation https://github.com/Nik-V9/U-Det, accessed on 11 March 2023.

## 5. Experimental Results

This section covers the details of the ablation study, overall performance of the proposed method, and experimental results.

### 5.1. Ablation Experiment

An ablation experiment was designed, based on the U-Net architecture with the LUNA16 test set, to verify the effectiveness of each component in the proposed architecture. The outcomes of this experiment are displayed in Table 3.

#### 5.1.1. Effect of Mish Activation Function

From Table 3, U-Net + Mish indicates the incorporation of the Mish activation function instead of the ReLU activation function of the original U-Net architecture. The DSC score of the original U-Net was 77.84%. It can be noticed that after implementing the Mish function in U-Net, this score was increased to 78.82%. On the other hand, an encoder consisting of the contraction path of the U-Net along with the Bi-FPN was implemented. On adding the Mish activation function to the abovementioned architecture, the DSC score became 80.22%. Moreover, a version of the proposed U-Det model with ReLU was applied, which performed marginally inferior to the Mish version. It can be observed that the percentage increase in performance due to Mish was nearly 1.3. Thus it is evident that the Mish activation function is useful in the U-Det model.

#### 5.1.2. Effect of Bi-FPN

In the description of Table 3, it can be discovered that Encoder + Bi-FPN replaced the backbone architecture with only a contraction path and the Bi-FPN functioned as the feature enricher and decoder. It can be observed that this architecture showed an improvement over the basic U-Net and achieved a DSC of 79.21%. Moreover, the ReLU version of the U-Det model is an incorporation of the Bi-FPN in the U-Net architecture; here, the DSC score showed 81.63%, which was a significant improvement over the original U-Net.

In addition to the above, even though the Bi-FPN is less computationally expensive than the U-Net architecture’s expansive path in terms of parameters, the Encoder + Bi-FPN successfully incorporates multiple feature fusions. The multi-feature fusion allows simultaneous feature map enhancement, thereby showing improvement over the U-Net architecture. In addition to that, the multiple implementations of Bi-FPN may serve as a decoder pathway, but it results in more complexity, computational expense and did not result in significant improvement in our study. Thus it can be inferred that the implementation of Bi-FPN between the expansive and contractive paths proved to be very effective in the proposed model.

#### 5.1.3. Effect of Bi-FPN + Expansion Path

The combination of Bi-FPN and the expansion path (ReLU version of the U-Det model) was much more productive compared to the Encoder + Bi-FPN; it exhibited a DSC score of 81.63%. The addition of the expansion path of U-Net to the Encoder + Bi-FPN model helped in the proper up-sampling of low-level features and a combination of feature maps from Bi-FPN, thereby enabling the efficient fusion of high-level semantic features with low-level features.

#### 5.1.4. Conclusion of the Ablation Study

From the observation of the DSC score of the U-Det model in Table 3 (82.82%), it is evident that the proposed U-Det model shows significant improvement over U-Net. Additionally, the effectiveness of all components and their culmination in the proposed model was verified through the ablation study.

### 5.2. Overall Performance

The histogram of the DSC values and the total amount of nodules, centered on every sample in the LUNA16 test set, is plotted in Figure 5 for better evaluation of the output of the U-Det model on the test set. From Figure 5, it can be quickly concluded that most of the nodules have a DSC value greater than 0.8.

For verifying the effectiveness of the Bi-FPN, the DSC results on the LUNA16 test set were compared with those of the original U-Net architecture. It can be observed that the U-Net model had a DSC of 77.84%, whereas the proposed model obtained a DSC of 82.82%, representing robust performance in the task of segmentation. Furthermore, by containing fewer parameters than the original U-Net architecture, the proposed U-Det model has shown its potential for efficient feature extraction and segmentation.

The segmentation results of complex cases, including attached (juxtapleural and juxta-vascular) and small-size nodules from both the LUNA16 and QIN Lung CT datasets, are shown in Figure 6. On observation of the results, it becomes apparent that the U-Det model outperforms the ground truth labels indicating the generalization potential of the model. The mean DSC outcomes on various lung nodule types from both datasets are shown in Table 4 and Table 5. By examining the experimental data shown in Table 4 and Table 5, it can be observed that the U-Det model’s potential for robust segmentation does not depend upon the type of nodule as it has shown exceptional performance on even small-size nodules.

## 6. Discussion

The following distinctions are noted when the proposed work is compared with similar other approaches.

To overcome the challenge of segmentation of nodules having small diameter and an intensity comparable to that of the surrounding noise, the proposed model used Bi-FPN, which functioned as a feature enricher, integrating multi-scale feature fusion for the purpose of efficient feature extraction.The proposed method applied a data augmentation technique to prevent the model from over-fitting and to obtain better segmentation results.The comparison of the proposed model with others showed high segmentation performance on small nodules and various other categories of pulmonary nodules.

For the experimental comparison, the results of the proposed method on LUNA16 dataset were compared with the results of methods reported in the literature such as MV-CNN [51], MCROI-CNN [23], MC-CNN [16], FCN-UNET [5], MV-DCNN [18], CF-CNN [27], and Cascaded-CNN [22]. It can be observed that the U-Det model performs comparably to human experts; the efficiency of the segmentation (in terms of DSC) of the four radiologists who worked on the LUNA16 is known to be 82.25%. Moreover, the proposed model was compared with models ranging from the original U-Net to various other recent convolution networks, including DDRN [30] and DB-ResNet [28].

In Table 6 and Table 7, and Figure 7 and Figure 8, the quantified results of the various methods on the LUNA16 test set and QIN Lung CT Segmentation dataset are represented. The outputs are shown as ’mean ± standard deviation’. As depicted in Table 6 and Table 7, the proposed method showed better performance than the existing segmentation methods on both datasets, indicating the robustness of the model.

Even though the MV-CNN [51], FCN-UNET [5], MCROI-CNN [23], MC-CNN [16], Cascaded-CNN [22], and CF-CNN [27] methods achieved good results, DDRN [30], DB-ResNet [28], and U-Det showed better performance compared to them. Although the DB-ResNet achieved good performance in various cases, its performance was hindered in the cases where the size of the nodule was less than 6 mm [28]. Similarly, for DDRN, the performance of the model was hindered when it was weakly supervised with the ground truth masks and also in the cases where the nodule size was small [30]. In contrast, the proposed method outperformed the ground truth segmentation masks and showed excellent generalization capability, as depicted in Figure 6. Further, Figure 9 illustrates the U-Det model’s performance in challenging cases such as small nodules (<6 mm), cavitary nodules, and juxta-vascular and juxtapleural nodules from the LUNA16 dataset. This observation confirmed that the proposed model showed efficient performance on various nodules, including nodules of less than 6 mm.

## 7. Conclusions

Lung CT image segmentation plays a vital role in the analysis of lung images, which helps in the identification of lung cancer. For lung nodule image segmentation, this paper proposed a deep-learning-based encoder–decoder model (U-Det) using Bi-FPN as a feature enricher by incorporating multi-scale feature fusion. The proposed method demonstrated encouraging precision in the segmentation of the lung nodules and obtained 82.82% and 81.66% DSC scores for the LUNA16 and QIN datasets, respectively. A comparison with similar approaches also showed the efficient performance of the proposed approach. It also achieved efficient segmentation results in daunting cases such as cavitary nodules, GGO nodules, juxtapleural nodules, and small nodules of less than 6 mm.

There are several areas of interest for further research. We employ a Bi-FPN to achieve a multi-scale fusion of features from 2D CT slices in the current work. However, it would be interesting to leverage the spatial information obtained from features of a voxel-based 3D CT scan representation. When fused across a multi-scale, these spatial features could potentially help models distinguish 3D properties and eccentricity of lung nodules for both segmentation and classification. Hence, future work will focus on developing a 3D capsule network based on the components of U-Det for fully automated malignancy classification and voxel-based segmentation of lung cancer.

## Figures and Tables

**Figure 1 diagnostics-13-01406-f001:**
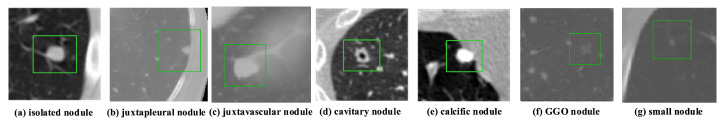
Illustrations of various types of lung nodules present in CT scans. Note: Small nodule indicates a nodule of diameter <6 mm. (**a**) isolated nodule; (**b**) juxtapleural nodule; (**c**) juxtavascular nodule; (**d**) cavitary nodule; (**e**) calcific nodule; (**f**) GGO nodule; (**g**) small nodule.

**Figure 2 diagnostics-13-01406-f002:**
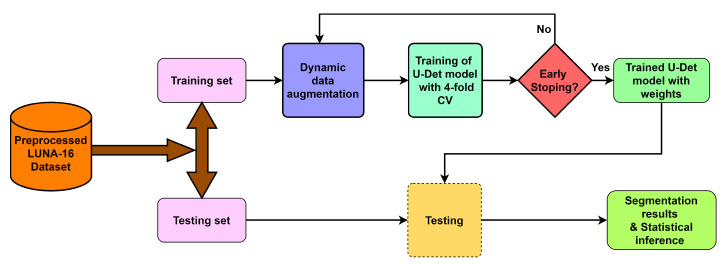
Overview of the proposed model pipeline.

**Figure 3 diagnostics-13-01406-f003:**
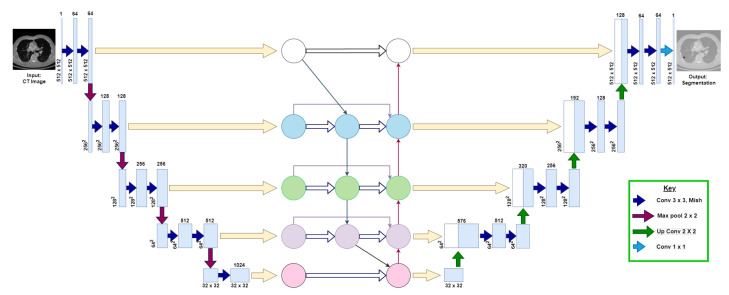
Illustration of the proposed U-Det model, where the convolutional neural network block between the down-sampling and up-sampling sections represents the Bi-FPN. The numbers at each layer of the architecture indicate the shape of feature maps at each layer, respectively. The key indicates the representation of various operations that take place in the backbone architecture.

**Figure 4 diagnostics-13-01406-f004:**
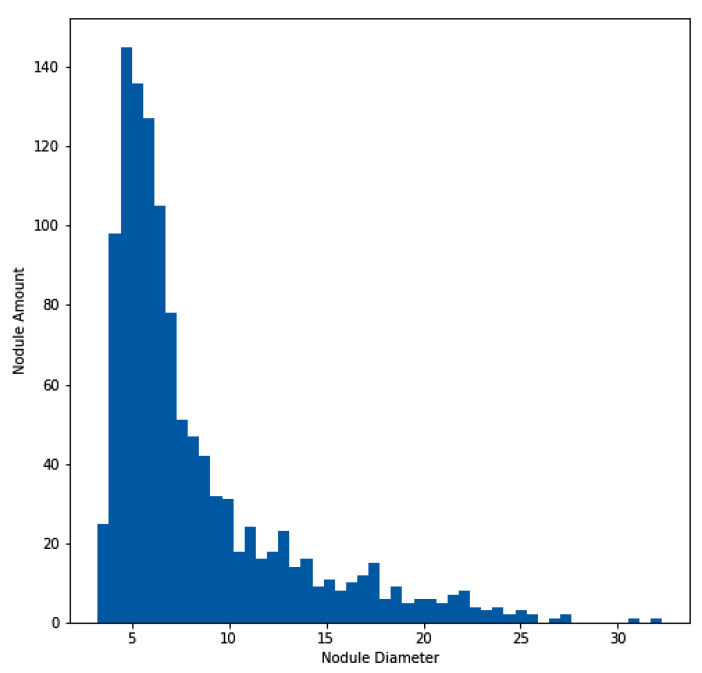
Histogram of lung nodule sizes across the LUNA16 dataset.

**Figure 5 diagnostics-13-01406-f005:**
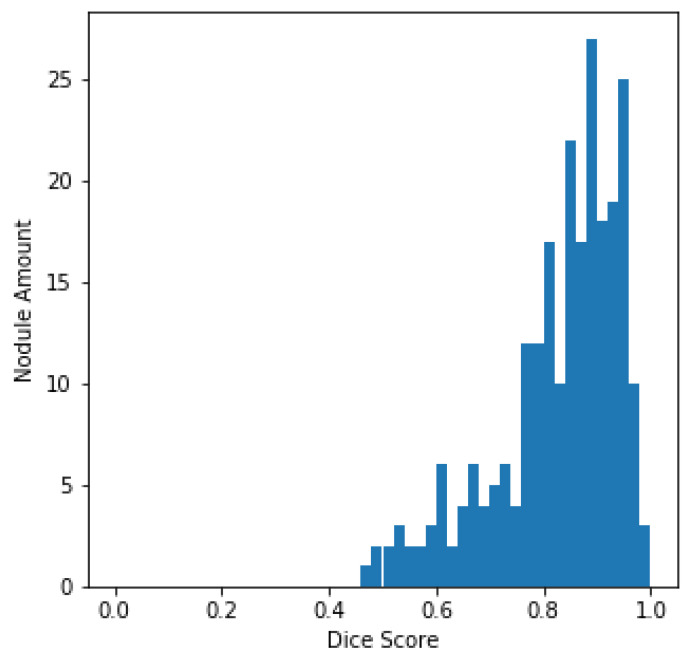
LUNA16 test set DSC distributions.

**Figure 6 diagnostics-13-01406-f006:**
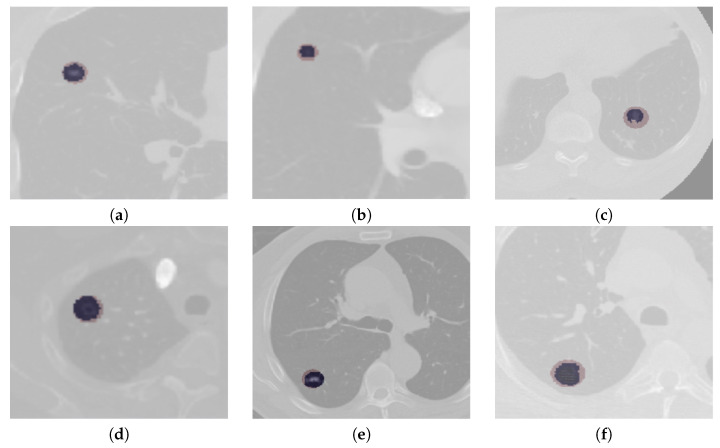
Qualitative depiction representing the ground truth masks and segmentation results of the U-Det model. Here, figures (**a**–**c**) are from the LUNA16 dataset, and (**d**–**f**) are from the QIN Lung CT Segmentation dataset. Note: The red filter represents the ground truth mask, and the blue filter represents the segmentation results.

**Figure 7 diagnostics-13-01406-f007:**
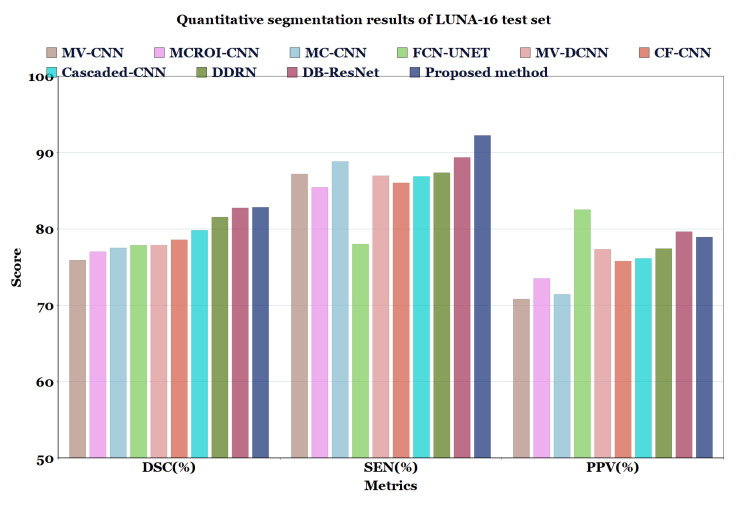
Quantitative segmentation results of LUNA-16 test set.

**Figure 8 diagnostics-13-01406-f008:**
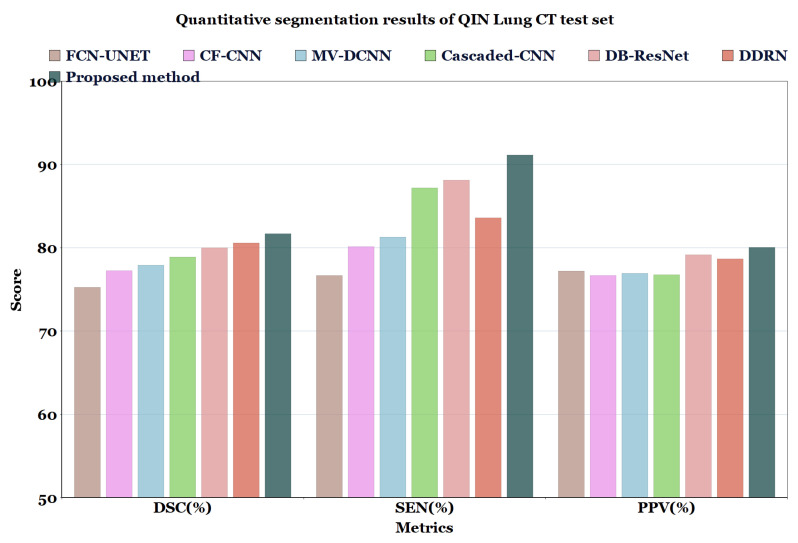
Quantitative segmentation results of QIN Lung CT test set.

**Figure 9 diagnostics-13-01406-f009:**
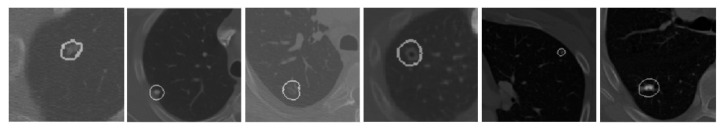
Visualization of segmentation results of the proposed U-Det model on heterogeneous types of lung nodules. The various types of lung nodules presented from left to right are isolated, juxtapleural, GGO and juxta-vascular, cavitary, very small size, and calcified.

**Table 1 diagnostics-13-01406-t001:** The layers and respective network parameters of the proposed model.

Layer Name	Number of Parameters
Contraction path:	
Conv2D × 10, Mish	1.884 $\times \;10^{7}$
MaxPool2D × 4	-
Bi-FPN:	
Conv2D × 5	1.269 $\times \;10^{5}$
BatchNormalization × 12	3072
ReLU × 12, MaxPool2D × 3	-
DepthwiseConv × 7	4032
Expansion path:	
Conv2D × 9, Mish	6.821 $\times \;10^{6}$
Conv2DTrans × 4, Mish	2.786 $\times \;10^{6}$
Total parameters:	2.858 $\times \;10^{7}$

**Table 2 diagnostics-13-01406-t002:** Distribution of LUNA16 train and test sets. The values are indicated as ’mean ± standard deviation’.

Characteristics	Train Set (*n* = 922)	Test Set (*n* = 244)
Diameter (mm)	8.13 ± 4.60	9.07 ± 5.24
Margin	4.03 ± 0.82	4.06 ± 0.76
Spiculation	1.60 ± 0.79	1.65 ± 0.87
Lobulation	1.73 ± 0.73	1.82 ± 0.80
Subtlety	3.91 ± 0.82	4.06 ± 0.78
Malignancy	2.95 ± 0.92	3.03 ± 1.00

Note: The range for all distinctive feature values except diameter was observed between 1 and 5. The ‘Margin’ characteristic shows nodule edge clarity. Both ‘Spiculation’ and ‘Lobulation’ indicate the shape characteristics of the nodule. ‘Subtlety’ explains the contrast between the nodule zone and its surrounding areas. ‘Malignancy’ reflects the possibility of this characteristic in a nodule.

**Table 3 diagnostics-13-01406-t003:** Ablation experiment on the LUNA16 test dataset. The experiment was based on the U-Net model.

Method	DSC (%)	SEN (%)	PPV (%)
U-Net	77.84 ± 21.74	77.98 ± 24.52	82.52 ± 21.53
U-Net + Mish	78.82 ± 22.01	78.97 ± 24.83	83.56 ± 21.80
Encoder + Bi-FPN	79.21 ± 12.49	84.40 ± 13.51	76.30 ± 14.42
Encoder + Bi-FPN + Mish	80.22 ± 12.33	85.47 ± 13.48	78.58 ± 14.34
Encoder–Decoder + Bi-FPN + ReLU	81.63 ± 11.85	91.06 ± 13.96	77.94 ± 13.68
Proposed Method	82.82 ± 11.71	92.24 ± 14.14	78.92 ± 17.52

**Table 4 diagnostics-13-01406-t004:** Segmentation results of the proposed model on the LUNA16 dataset for various cases such as attached and non-attached nodules, and nodules of large and small sizes.

**(a) LUNA16 Test set**				
	Attached (*n* = 56)	Non-Attached (*n* = 188)	Diameter < 6 mm (*n* = 104)	Diameter ≥ 6 mm (*n* = 140)
**DSC (%)**	81.82	83.11	83.40	82.40

**Table 5 diagnostics-13-01406-t005:** Segmentation results of the proposed model on the QIN Lung CT dataset for various cases such as attached and non-attached nodules, and nodules of large and small sizes.

**(b) QIN Lung CT Segmentation dataset**				
	Attached (*n* = 34)	Non-Attached (*n* = 122)	Diameter < 6 mm (*n* = 54)	Diameter ≥ 6 mm (*n* = 102)
**DSC (%)**	80.02	83.30	83.10	80.22

**Table 6 diagnostics-13-01406-t006:** Quantitative segmentation results of the proposed model compared to different types of model architectures on LUNA16 dataset.

LUNA16 Test Set			
**Network Architecture**	**DSC (%)**	**SEN (%)**	**PPV (%)**
MV-CNN [51]	75.89 ± 12.99	87.16 ± 12.91	70.81 ± 17.57
MCROI-CNN [23]	77.01 ± 12.93	85.43 ± 15.97	73.52 ± 14.62
MC-CNN [16]	77.51 ± 11.4	88.83 ± 12.34	71.42 ± 14.78
FCN-UNET [5]	77.84 ± 21.74	77.98 ± 24.52	82.52 ± 21.53
MV-DCNN [18]	77.85 ± 12.94	86.96 ± 15.73	77.33 ± 13.26
CF-CNN [27]	78.55 ± 12.49	86.01 ± 15.22	75.79 ± 14.73
Cascaded-CNN [22]	79.83 ± 10.91	86.86 ± 13.35	76.14 ± 13.46
DDRN [30]	81.56 ± 11.59	87.35 ± 12.39	77.42 ± 14.65
DB-ResNet [28]	82.74 ± 10.19	89.35 ± 11.79	79.64 ± 13.54
Proposed Method	82.82 ± 11.71	92.24 ± 14.14	78.92 ± 17.52

**Table 7 diagnostics-13-01406-t007:** Quantitative segmentation results of the proposed model compared to different types of model architectures on QIN Lung CT dataset.

QIN Lung CT Segmentation Dataset			
**Network Architecture**	**DSC (%)**	**SEN (%)**	**PPV (%)**
FCN-UNET	75.26 ± 11.82	76.65 ± 16.42	77.21 ± 11.57
CF-CNN	77.23 ± 11.53	80.12 ± 17.07	76.65 ± 12.20
MV-DCNN	77.89 ± 10.64	81.29 ± 15.60	76.95 ± 11.62
Cascaded-CNN	78.89 ± 11.89	87.20 ± 12.44	76.74 ± 13.52
DB-ResNet	80.01 ± 11.46	88.13 ± 12.34	79.13 ± 14.12
DDRN	80.56 ± 11.08	83.57 ± 11.78	78.65 ± 13.86
Proposed Method	81.66 ± 10.09	91.11 ± 12.01	80.05 ± 13.34

## Data Availability

More details can be found at https://github.com/Nik-V9/U-Det.

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
