# Peer review of "A Bi-FPN-Based Encoder–Decoder Model for Lung Nodule Image Segmentation"

_diagnostics, 2023, doi:10.3390/diagnostics13081406_

Round 1

Reviewer 1 Report

Thank you for giving me the opportunity to review this manusctipt. Overall, the manuscript is well-written and provides a detailed description of a deep learning model for nodule segmentation. However, I have few recommendations to the authors on few areas where the manuscript could be improved. 

1. The manuscript includes several technical terms and concepts that may not be familiar to all readers. Providing more detailed explanations, perhaps a supplemental mateiral, of these terms can help readers to better understand the methodology and results of the study.

2. In the Methodology section, the authors have described the U-Det model architecture and the Bi-FPN module in detail. However, they could expand more about the rationale behind their design choices and how these choices help in improving the segmentation accuracy. And also, I think the authors could expand more on the limitations of the study. 

3. It would be benificial to the readers if the authors could provide the source code or a link to the implementation of their approach, to facilitate the replication of their results by other researchers.

Author Response

We appreciate the time and efforts made by the editor and reviewers while reviewing this manuscript. We pay our sincere thanks to the esteemed reviewers for their valuable comments and suggestions. We hope that we have addressed all the issues raised by the reviewers. The changes made in the revised manuscript are highlighted in BLUE color. The detailed responses for incorporating the reviewer’s comments are attached as a file.

Reviewer 2 Report

"The article's abstract does not mention the network's training parameters. Please provide this information."

"To support the claim that many modified U-Net architectures have achieved significant results in different domains of biomedical imaging, appropriate references  should be cited."

"To support the claim that CNN architectures for lung nodule segmentation are still immature, appropriate references  should be cited."

"Why wasn't the article's discussion of segmentation methods based on image fusion mentioned?"

"While Figure 2 provides a flowchart of the proposed model, more details are needed to fully explain it."

"Why weren't other measurement metrics for segmentation, such as IoU, used in the study?"

"Is there a link or GitHub repository where the dataset used in this study can be accessed?"

"Is there a GitHub link to access the code used in this study?"

Author Response

(The authors gave the same response as above.)
